# Interactive Brain-Tumor Analysis with Model Agnostic Hybrid Augmentation

## Abstract

Brain tumor segmentation and survivability prediction are crucial in neuro-oncology, directly impacting clinical decision-making and patient management. However, traditional deep learning-based segmentation approaches often lack flexibility, interpretability, and adaptability to user-driven corrections, limiting their clinical utility. To overcome these challenges, we introduce RadBot, a novel vision-language model (VLM)-powered framework that integrates tumor segmentation, interpretative analysis, and survivability prediction into a unified, interactive pipeline. Moreover, prompt-based VLMs, such as CLIPSeg, exhibit sensitivity to linguistic variations inherent in English prompts, which often fail to span the full vector embedding range for nuanced tumor morphologies. To mitigate prompt sensitivity without retraining, we introduce model agnostic hybrid augmentation (MAHA), an inference-time prompt ensemble method for brain tumor analysis. To improve interpretability, we incorporate LLaVA, a multimodal large language model (MMLLM), enabling interactive question-answering for tumor analysis. Additionally, the RadBot Mask Editor provides an interactive refinement tool, allowing radiologists to manually correct segmentation errors through brushing and unbrushing tools, ensuring clinically precise results. For survivability prediction, RadBot integrates LLaVA-based analysis of MRI and clinical data for efficient prognosis estimation and decision-support. We validate proposed Rad-Bot+MAHA on BraTS 2020 and 2021 datasets, achieving SOTA segmentation performance. Our findings demonstrate that integrating VLMs and MMLLMs enhances segmentation accuracy, interpretability, and clinical relevance. RadBot bridges the gap between automated segmentation and expert-driven analysis, establishing a new paradigm for AI-assisted workflows.

## 1 Introduction

Brain tumor segmentation and survival prediction (Anand et al., 2021) are critical tasks in neuro-oncology that directly impact treatment planning and patient outcomes. The accurate delineation of tumor regions from multimodal Magnetic Resonance Imaging (MRI) scans, enables clinicians to make informed decisions about treatment strategies. However, these tasks present significant challenges due to the heterogeneous appearance of brain tumors, the complexity of multimodal MRI data, and the need to integrate clinical information for comprehensive analysis.

Annotation of brain tumor regions in MRI scans is a time-intensive and laborious task for radiologists. Accurate segmentation requires meticulous effort to delineate tumor boundaries across multiple slices and modalities, which can be both physically and cognitively demanding. While convolutional neural networks (CNNs), generative adversarial networks (GANs) (Mukherkjee et al., 2022; Nema et al., 2020), transformers (Cao et al., 2022; Du et al., 2022), and ensemble-based methods (Phophalia & Maji, 2017; Győrfi et al., 2019; Hossain et al., 2023) have shown promise in automating this process, they often lack the flexibility to accommodate user-driven corrections. This limitation can be problematic in clinical workflows, where even minor inaccuracies in segmentation may lead to suboptimal treatment planning. Moreover, existing methods (Mukherkjee et al., 2022; Nema et al., 2020; Cao et al., 2022; Du et al., 2022; Phophalia & Maji, 2017; Győrfi et al., 2019; Hossain et al., 2023) typically treat segmentation as a static process, offering little to no support for post-prediction editing. These approaches fail to provide interpretability or interactivity, leaving radiologists unable to refine the results to match clinical expectations. The inability to edit segmentation masks directly within the pipeline creates a gap between automated tools and real-world

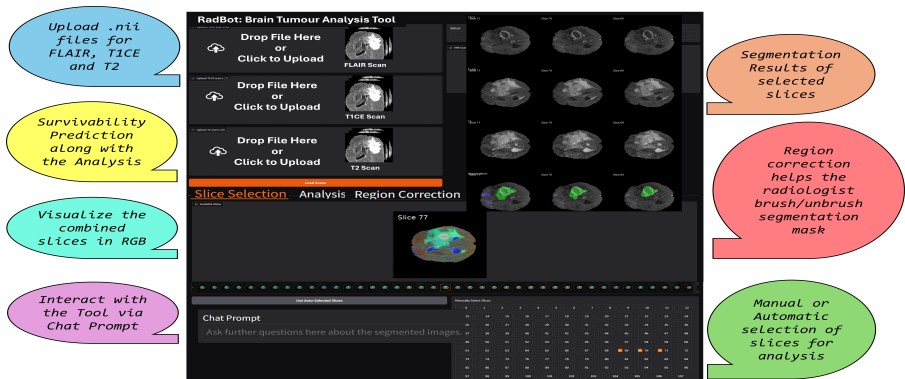

Figure 1: RadBot – brain tumor segmentation and radiological analysis tool. The system leverages MRI scans (FLAIR, T1CE, and T2) to assist in brain tumor analysis via automated or manually selected slice processing. It delineates whole tumor, tumor core, and enhancing tumor, supports visual question answering (VQA) for tumor insights, and offers interactive mask editing (brushing/unbrushing) for clinical usability.

usability. Addressing this challenge requires a framework that not only delivers accurate segmentation but also empowers users with the ability to interactively modify and refine the results, ensuring both precision and clinical relevance.

Recent advances in vision-language models (VLMs) (Agarwal et al., 2021) and multimodal large language models (MMLLMs) (Liu et al., 2023a) have shown promising results in medical image analysis. These models offer unique advantages in terms of interpretability and flexibility through their ability to process both visual and textual information. However, their potential for brain tumor analysis, particularly in combining segmentation with survival prediction, remains largely unexplored. Additionally, VLMs like CLIPSeg (Lüddecke & Ecker, 2022; Chen et al., 2024) are *prompt-sensitive*: a single English instruction often under-specifies heterogeneous tumor appearance, especially in low-contrast ET and TC.

To surmount these hurdles, we propose **RadBot**, a pioneering interactive framework that orchestrates VLM-driven segmentation, MMLLM-powered analysis, and survival prediction into a cohesive, clinician-centric pipeline. At its core, RadBot harnesses CLIPSeg (Lüddecke & Ecker, 2022) for multimodal feature extraction and three-channel mask generation (whole tumor, tumor core, enhancing tumor). To conquer the brittleness of single-prompt conditioning-stemming from English's semantic ambiguities that compress task context into narrow embeddings (Chen et al., 2024), we introduce Model-Agnostic Hybrid Augmentation (**MAHA**) approach, an inference-time ensemble that orchestrates diverse natural language prompts (e.g., variations on "delineate enhancing tumor core") via threshold-aware logits fusion. The main contributions of the proposed work are as follows:

- **RadBot:** An interactive, deployable multimodal framework enabling radiologists to generate segmented reports and inspect each pipeline step for transparency and interpretability (see Section 3.1).
- **Unified pipeline:** A single, streamlined system that performs segmentation (WT, TC, ET) and downstream analysis, improving efficiency over cascaded or ensemble approaches (see Section 3.2).
- **RadBot Mask Editor:** An interactive editor to correct false positives/negatives with brushing and erasing, adjustable brush sizes, continuous painting, and coordinate-precise edits (see Section 4.5).
- **MAHA:** A hybrid prompting strategy that improves tumor segmentation by combining multiple instructions through logit-level fusion and adaptive thresholding. It works entirely at inference time, requires no retraining, and is compatible with any vision-language model (see Section 3.5).

We conduct extensive experiments and ablation studies using state-of-the-art VLMs and MMLLMs. Our framework is evaluated on the benchmark BraTS 2020 and 2021 datasets, demonstrating its effectiveness for both brain tumor segmentation and survivability prediction. Figure 1 summarises the main contributions of RadBot. below.

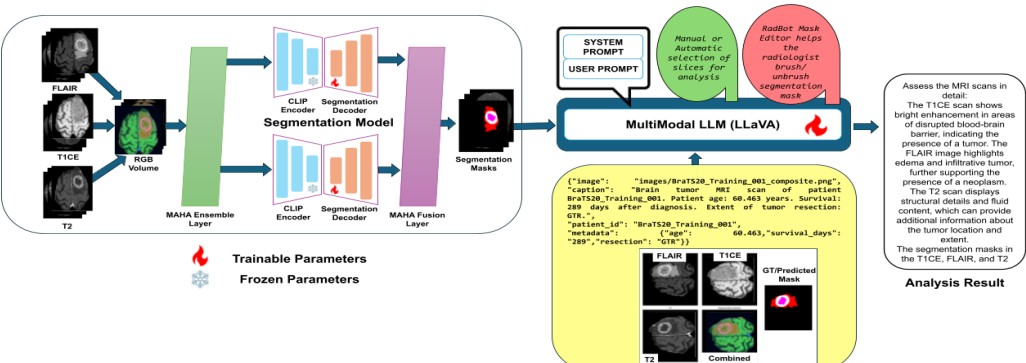

Figure 2: Overview of RadBot for brain tumor segmentation and radiological analysis. Given MRI scans (FLAIR, T1CE, and T2) and a textual prompt, RadBot extracts visual and textual features using a frozen CLIP encoder. The CLIPSeg decoder generates tumor segmentation masks, which are further enhanced by **MAHA**, an inference-time prompt ensembling method that improves segmentation accuracy on subtle regions. The interactive mask editor enables manual refinements. LLaVA-based analysis provides clinical insights, while a dual-branch model integrates imaging and clinical data for survivability prediction, supporting radiological assessment and decision-making.

## 2 RELATED WORK

Brain tumor segmentation is a critical task in medical imaging, with significant advancements driven by deep learning and multimodal approaches. In this section, we review prior work in brain tumor segmentation, vision–language models, and multimodal large language models.

Brain tumor segmentation has been extensively studied, with the BraTS challenges (Menze et al., 2014; Bakas et al., 2018) serving as a benchmark for evaluating segmentation algorithms. The advent of deep learning, particularly convolutional neural networks (CNNs), revolutionized the field, with U-Net (Ronneberger et al., 2015) and its variants (Isensee et al., 2021) becoming widely adopted for medical image segmentation due to their encoder–decoder architecture. Recent advances in brain tumor segmentation have explored various architectural innovations. Isensee et al. (Isensee et al., 2021; 2024) proposed nnU-Net, a self-configuring framework that automates key aspects of the segmentation pipeline including preprocessing, network architecture, and training strategy. This approach adapts to dataset-specific characteristics, making it widely applicable across various segmentation tasks. Jia et al. (Jia et al., 2021) introduced H2NF-Net, leveraging hierarchical feature extraction paired with non-local attention mechanisms to better capture spatial dependencies in multimodal MRI data, securing second place in the BraTS 2020 Segmentation Challenge (Mehta et al., 2022). Fidon et al. (Fidon et al., 2018; 2021) advanced a framework employing the Generalized Wasserstein Dice Score with distributionally robust deep learning to better handle rare tumor classes and mitigate class imbalance. Raza et al. (Raza et al., 2023) introduced DRESU-Net, a 3D deep residual U-Net model that leverages residual connections to enhance feature propagation through the network, facilitating the training of deeper architectures for multimodal MRI processing. Liu et al. (Li et al., 2022) proposed a multiscale lightweight 3D segmentation algorithm incorporating attention mechanisms within a computationally efficient framework, enhancing feature representation at various resolutions while balancing real-time processing requirements with high segmentation accuracy. Schwehr and Achanta (Schwehr & Achanta, 2025) developed a region-of-interest detection algorithm implemented during data preprocessing to locate salient features and remove extraneous MRI data, decreasing input size and allowing for more aggressive data augmentations and deeper neural networks. Despite these advances, existing methods (Mukherkjee et al., 2022; Nema et al., 2020; Cao et al., 2022; Du et al., 2022; Phophalia & Maji, 2017; Győrfi et al., 2019; Hossain et al., 2023) often face challenges related to computational complexity, interpretability, and integration of multimodal data. Further, these approaches rely on cascaded or ensemble models, which can be difficult to deploy in clinical settings due to their complexity and resource requirements.

Vision–language models (VLMs) such as CLIP (Agarwal et al., 2021) have demonstrated strong performance in general image understanding by leveraging large-scale pretraining on paired image–text datasets. The extension of VLMs to medical imaging has shown promise, particularly in tasks requiring multimodal reasoning. For instance, CLIP-based models have been adapted for medical

image classification and segmentation, as demonstrated by Lüddecke et al. (Lüddecke & Ecker, 2022), who introduced CLIPSeg, a model that combines CLIP's visual-textual understanding with a segmentation-specific decoder. In the context of brain tumor segmentation, VLMs offer unique advantages by incorporating textual prompts to guide the segmentation process. This approach enhances model interpretability and allows for flexible adaptation to different tasks. However, the application of VLMs in medical imaging remains underexplored, with limited studies addressing their integration with clinical data for comprehensive analysis.

Recent works have further explored prompt engineering and ensembling to mitigate the brittleness of single-prompt conditioning, where linguistic variations lead to inconsistent embeddings and sub-optimal delineation of subtle tumor subregions like the enhancing tumor (ET) and tumor core (TC). For example, Avogaro et al. (Avogaro et al., 2025) proposed PromptMatcher, a training-free fusion of text and visual prompts for few-shot medical segmentation, achieving strong zero-shot performance. Similarly, VLSM-Ensemble (Dietlmeier, 2025) ensembles multiple CLIP-based VLMs at the feature level to boost robustness in medical segmentation. Building on these foundations, our MAHA advances inference-time prompt ensembling by orchestrating diverse natural language prompts (e.g., semantic variations like "delineate necrotic core" vs. "outline enhancing rim") and fusing their probability maps via adaptive thresholding. Unlike PromptMatcher's (Avogaro et al., 2025) static alignment or VLSM-Ensemble's (Dietlmeier, 2025) model-level stacking, MAHA dynamically expands the embedding manifold-drawing from latent-space diversity principles to handle low-contrast heterogeneities.

Survivability prediction in brain tumor patients is a challenging task that requires the integration of imaging biomarkers and clinical data. Traditional approaches relied on statistical models and handcrafted features, which often failed to capture the complex interactions between imaging and clinical variables. Deep learning-based methods have shown significant improvements by leveraging CNNs for feature extraction from MRI scans. For example, Pei et al. (Pei et al., 2020) proposed a CNN-based model for survival prediction, demonstrating the importance of imaging biomarkers in prognostic modeling. Multimodal approaches that combine imaging and clinical data have further improved survivability prediction. For example, Chaddad et al. (Chaddad et al., 2019) introduced a framework that incorporates not only radiomic features but also clinical variables to achieve state-of-the-art performance. However, these methods often lack scalability and generalizability due to the small size of medical datasets.

Multimodal large language models (MMLLMs) such as LLaVA (Liu et al., 2023a), GPT-4 (Achiam et al., 2023), GPT-4o (Hurst et al., 2024), and DeepSeek (Liu et al., 2024) have recently emerged as powerful tools for vision–language analysis. These models combine the capabilities of vision and language understanding, enabling them to generate detailed insights from medical images. LLaVA, in particular, has demonstrated strong performance in medical imaging tasks, including segmentation and descriptive analysis. Despite significant progress in brain tumor segmentation, existing approaches face challenges in computational complexity, interpretability, and integration of multimodal data. Our proposed RadBot addresses these limitations via VLM-driven segmentation and MMLLM-based interpretative analysis, constituting a complete and efficient solution for brain tumor analysis.

## 3 PROPOSED RADBOT

Our proposed framework, RadBot, integrates vision–language models for brain tumor segmentation with a subsequent survivability analysis, leveraging both imaging and clinical data. The framework processes MRI slices, generates segmentation masks, and performs downstream analysis using large vision–language models (Figure 2).

### 3.1 TUMOR SEGMENTATION VLM ARCHITECTURE

We utilize the CLIPSeg architecture (Lüddecke & Ecker, 2022) for brain tumor segmentation, which combines the visual understanding capabilities of CLIP with a decoder specialized for segmentation tasks. The model consists of a CLIP encoder that processes both image and text inputs, followed by a decoder that produces segmentation masks.

In our implementation, we modify the decoder to output *three* segmentation channels corresponding to distinct tumor regions, rather than the original single-channel output. Specifically, we replace the final transposed convolution layer with a new layer that maintains the input channel dimension of

32 but expands the output to three channels. Kernel size and stride are preserved at $4 \times 4$ and $4 \times 4$, respectively, ensuring that output spatial dimensions remain consistent with the original architecture.

Weights of the new output layer are initialized with Xavier uniform initialization to promote stable optimization. During training, we freeze the entire CLIP encoder (image and text branches) to preserve its large-scale pretraining representations. Only the **1.1**M decoder parameters remain trainable, allowing task-specific adaptation for brain tumor segmentation while leveraging robust frozen features.

This approach substantially reduces the number of trainable parameters versus end-to-end training, improving efficiency and mitigating overfitting risks on modest-sized medical datasets.

## 3.2 CLINICAL DATA HANDLING

The input to our segmentation model consists of three MRI modalities-FLAIR, T1CE, and T2 processed as separate channels. These provide complementary tissue cues: FLAIR highlights edema, T1CE emphasizes the enhancing tumor core via contrast, and T2 delineates overall tumor extent. Each MRI slice is preprocessed and resized to $128 \times 128$. Segmentation is conditioned by a textual prompt ("*An MRI scan of the brain. Identify the tumor.*"), which is encoded using the CLIP text encoder. The decoder produces three output channels corresponding to segmentation masks: Channel 0 represents the non-enhancing tumor core, Channel 1 corresponds to the peritumoral edema, and Channel 2 denotes the enhancing tumor core. From these outputs, we compute clinically relevant aggregates that are used for evaluation and downstream analysis: *Whole Tumor* (WT), defined as the union of all three channels; *Tumor Core* (TC), defined as the union of channels 0 and 2; and *Enhancing Tumor* (ET), corresponding directly to channel 2.

## 3.3 MULTIMODAL LLM

For interpretative analysis, we evaluated several multimodal large language models (LLMs), including LLaVA (Liu et al., 2023a), GPT-4 (Achiam et al., 2023), GPT-4o (Hurst et al., 2024), and DeepSeek (Liu et al., 2024). We selected LLaVA for its strong performance on medical image understanding and rich descriptive outputs.

We design a structured system prompt for brain tumor reporting that instructs the model to act as a radiology/neuro-oncology expert. Inputs to LLaVA are grid images that juxtapose original MRI slices with segmentation masks for WT, TC, and ET, enabling context-aware reasoning over both raw imaging and model outputs. This arrangement supports detailed, clinically relevant narratives aligned with radiologists' workflows and helps bridge automated segmentation with expert interpretation.

## 3.4 SURVIVABILITY PREDICTION USING LLAVA

To extend RadBot beyond segmentation and interpretative analysis, we develop a survivability prediction pipeline using LLaVA. We leverage BraTS 2020 survival metadata (survival days and extent of resection; 236 patients) to form image–caption pairs: composite grid images (T1CE, FLAIR, T2, plus ground-truth mask) with structured captions containing age, survival days, and resection extent.

This pairing allows the model to learn associations between imaging patterns and outcomes. We fine-tune with Low-Rank Adaptation (LoRA) (Hu et al., 2022), updating only a small subset of parameters so the base model retains general knowledge while adapting to survivability prediction. This yields accurate predictions without heavy compute and facilitates practical deployment.

By integrating survivability prediction, RadBot offers a unified framework for segmentation, interpretation, and prognosis, supporting informed clinical decision-making.

## 3.5 MODEL AGNOSTIC HYBRID AUGMENTATION (MAHA)

To enhance segmentation without retraining, we introduce **MAHA** as an inference-time module. MAHA ensembles predictions from multiple natural-language prompts via threshold-aware probability fusion (mean, max, product, and soft-logic variants). This improves the performance across WT/ET/TC while remaining compatible with any prompt-conditioned VLM. Because MAHA operates entirely at inference time, it provides gains with negligible engineering overhead. MAHA supports: mean (balanced), max (recall-oriented), product (soft-AND; precision-oriented), and a soft-logic rule that upgrades voxels confident under one prompt and near-threshold under another.

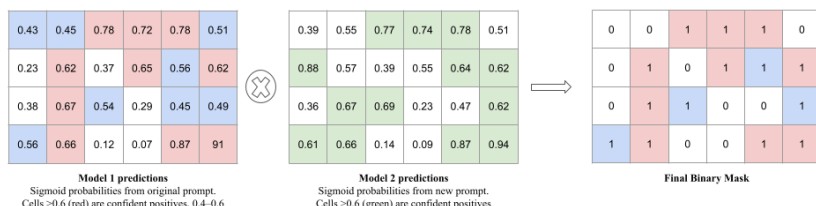

Figure 3: Illustration of the soft logic fusion rule. Left: predictions from Model 1 with high threshold $0.6$ (red) and low threshold $0.4$ (blue). Middle: predictions from Model 2 with high threshold $0.6$ (green). Right: final binary mask. Pixels above $0.6$ are kept directly; blue entries are rescued by the soft logic rule when Model 1 is in the $0.4$–$0.6$ range and Model 2 exceeds $0.6$.

Thresholds are swept on validation and fixed per region (WT, TC, ET). Binary masks are then composed into clinical aggregates for downstream analysis. For each prompt, MAHAM first derives a probability map by applying a sigmoid function to the model logits. These maps can then be combined through four different post hoc fusion strategies, each offering distinct trade-offs, and all computable efficiently on CPU. In *Mean Fusion*, the probability maps are averaged, which smooths out disagreements between prompts and generally results in strong overall Dice performance. *Max Fusion* instead takes the voxel-wise maximum across maps; this aggressive approach tends to favor recall, but may also increase false positives. *Product Fusion (Soft-AND)* computes the element-wise product of maps, strongly penalizing disagreements and thereby boosting precision. Finally, a *Soft Logic Rule* is applied in which

$$\hat{Y} = (P_1 > \tau) \ \vee \ \big((P_1 > \tau - \delta) \wedge (P_2 > \tau)\big), \tag{1}$$

where $\tau$ is the decision threshold and $\delta$ allows leeway for voxels that are close to the threshold. This rule balances strict thresholding with tolerance for borderline cases, improving robustness in ambiguous regions. Figure 3 illustrates the soft logic fusion rule with two prediction models.

**Advantages:** The proposed approach offers several advantages: it is plug-and-play, making it applicable to any prompt-based segmentation model; incurs zero training cost since it is inference-only with no gradient updates; and remains flexible, allowing prompts to be added or removed depending on the use case. Moreover, it is model-agnostic, functioning across different VLM backbones, and clinically meaningful, particularly in improving the segmentation of small or low-contrast ET and TC regions.

## 4 EXPERIMENTAL ANALYSIS

To evaluate the effectiveness of our proposed RadBot, we conducted an extensive experimental analysis for brain tumor segmentation, along with survivability prediction.

### 4.1 MRI DATA ACQUISITION

The proposed framework is trained on the BraTS 2020 challenge dataset (Mehta et al., 2022), which is divided into three cohorts: Training, Validation, and Testing. The Training dataset consists of multi-parametric MRI (mpMRI) scans from 369 diffuse glioma patients, split into 80 for training and 20 for testing, while the Validation dataset includes 125 cases. Each mpMRI scan comprises four imaging sequences: native T1-weighted (T1) for anatomical structure visualization, post-contrast T1-weighted (T1ce) for highlighting the active tumor region, T2-weighted (T2) for tumor core visibility excluding edema, and FLAIR (Fluid-Attenuated Inversion Recovery) for depicting the entire tumor region, including edema, non-enhancing core, enhancing core, and necrotic or fluid-filled areas. All MRI volumes undergo skull-stripping (brain extraction), co-alignment to a standard anatomical atlas (SRI24), and resampling to a uniform $1 \times 1 \times 1 \, \mathrm{mm}^3$ voxel resolution. Each scan has an image size of $240 \times 240 \times 155$, comprising 155 brain slices per patient.

### 4.2 TRAINING PIPELINE

Our training pipeline was designed to efficiently train the CLIPSeg model for brain tumor segmentation while maintaining computational feasibility. All experiments were conducted on an NVIDIA A6000 GPU (48 GB). We employed AdamW (Loshchilov & Hutter, 2017) with weight

Figure 4: Brain tumor segmentation results for MRI scans from the BraTS 2020 database for whole, core, and enhancing tumor regions. Seven sample MRI scans (FLAIR, T1CE, T2, and combined) and ground truth (GT) with their respective RadBot predictions. Note: red, white, and pink denote whole, enhancing, and core tumor regions, respectively.

Table 1: Performance comparison between RadBot and existing state-of-the-art methods for WT, TC, and ET segmentation (Dice Similarity Coefficient) on BraTS 2020.

| Method | WT | TC | ET | Mean |
|---|---|---|---|---|
| nnU-Net (Isensee et al., 2021) | 91.07 | 87.97 | 81.37 | 86.80 |
| H2NF-Net (Jia et al., 2021) | 91.30 | 85.50 | 78.80 | 85.20 |
| nnU-Net Ensemble (Fidon et al., 2021) | 91.00 | 84.40 | 77.60 | 84.33 |
| dResU-Net (Raza et al., 2023) | 86.60 | 83.57 | 80.04 | 83.40 |
| ADHDC-Net (Liu et al., 2023b) | 89.75 | 83.31 | 78.01 | 83.69 |
| TransBTS (Wang et al., 2021) | 90.09 | 81.73 | 78.73 | 83.51 |
| DR-Unet104 (Colman et al., 2021) | 86.73 | 79.06 | 75.14 | 80.31 |
| **RadBot (DeepLabV3+) (ours)** | 85.92 | 82.49 | 76.05 | 81.48 |
| **RadBot (CLIPSeg)(ours)** | 89.94 | 89.25 | 83.65 | 87.59 |
| **RadBot(CLIPSeg)+MAHA(ours)** | **92.15** | **92.24** | **85.18** | **89.86** |

decay $1 \times 10^{-2}$ and initial learning rate $1 \times 10^{-4}$. The loss function is Binary Cross-Entropy with Logits (BCEWithLogitsLoss). For segmentation, we compute Dice Similarity Coefficient (DSC) for WT, TC, and ET.

### 4.3 Brain Tumor Segmentation Results

We first evaluated our CLIPSeg-based vision–language model (VLM) on the BraTS 2020 dataset (Mehta et al., 2022), a widely used benchmark for brain tumor segmentation. The quantitative results are summarized in Table 1, where we report the Dice Similarity Coefficient (DSC) for the three tumor sub-regions: Whole Tumor (WT), Tumor Core (TC), and Enhancing Tumor (ET), along with the mean score. Our model achieves strong performance across all regions, attaining DSC values of 92.15 for WT, 92.24 for TC, and 85.18 for ET, resulting in a mean DSC of 89.86. These results indicate that the model is particularly effective in capturing both large-scale tumor structures (WT) and clinically critical subregions (TC and ET). Figure 4 illustrates representative qualitative results, while additional visualizations are provided in the supplementary material for completeness.

Further, we evaluated the RadBot framework on the BraTS 2021 dataset (Baid et al., 2021). In this setting, the model was fine-tuned on 80% of the training set and evaluated on the remaining 20%. Table 2 reports the comparative performance with respect to SOTA methods. RadBot+MAHA demonstrates competitive DSC, confirming the robustness of our pipeline. Moreover, when combined with the proposed MAHA fusion strategy, the RadBot+MAHA configuration attains a DSC of 92.01, thereby surpassing the previously reported SOTA result of 91.50. This improvement highlights the effectiveness of the proposed hybrid augmentation and fusion mechanism in enhancing segmentation performance for challenging tumor.

### 4.4 Ablation Study

We compare two VLM variants: CLIPSeg (Lüddecke & Ecker, 2022) and DeepLabV3+ (CLIP decoder) (Chen et al., 2018), both on BraTS 2020 (Mehta et al., 2022). DSC results (Table 1) show

Table 2: Performance comparison between the proposed RadBot and existing state-of-the-art methods for segmentation in terms of Dice Similarity Coefficient (DSC) on BraTS 2021 database.

| Method | DSC | Method | DSC |
|---|---|---|---|
| nnU-Net (org.) (Isensee et al., 2021) | 91.24 | SwinUNETR-V2 (He et al., 2023) | 90.74 |
| nnU-Net RsEnM (Isensee et al., 2024) | 91.26 | nnFormer (Zhou et al., 2021) | 90.22 |
| nnU-Net RsEnL (Isensee et al., 2024) | 91.13 | CoTr (Xie et al., 2021) | 90.73 |
| nnU-Net RsEnXL (Isensee et al., 2024) | 91.18 | Mamba Base (Ma et al., 2024) | 91.26 |
| MedNeXtLk3 (Roy et al., 2023) | 91.35 | U-MambaBot (Ma et al., 2024) | 91.26 |
| MedNeXtLk5 (Roy et al., 2023) | 91.50 | U-MambaEnc (Ma et al., 2024) | 90.91 |
| STU-NetS (Huang et al., 2023) | 90.55 | SwinUNETR (Tang et al., 2022) | 90.68 |
| STU-NetB (Huang et al., 2023) | 90.85 | **RadBot (Ours)** | 90.61 |
| STU-NetL (Huang et al., 2023) | 91.26 | **RadBot+MAHA (Ours)** | **92.01** |

Table 3: Impact of various segmentation mask thresholds for CLIPSeg and DeepLabV3+ (Dice) on BraTS 2020.

| Model | Threshold | WT | TC | ET |
|---|---|---|---|---|
| CLIPSeg | 0.6 | **89.94** | **89.25** | **83.65** |
| CLIPSeg | 0.5 | 82.73 | 80.92 | 76.82 |
| CLIPSeg | 0.7 | 86.96 | 85.82 | 81.35 |
| CLIPSeg | 0.8 | 82.73 | 80.92 | 76.82 |
| DeepLabV3+ | 0.5 | 81.92 | 81.40 | 75.60 |
| DeepLabV3+ | 0.6 | 85.92 | 82.49 | 76.05 |

Table 4: Ablation of MAHA with various fusion strategies in terms of DSC on BraTS 2020.

| Strategy | WT | ET | TC | Mean |
|---|---|---|---|---|
| RadBot (w/o MAHA) | 89.94 | 83.65 | 89.25 | 87.59 |
| MAHA-Max | 86.29 | 82.28 | 91.01 | 86.52 |
| MAHA-Product | 88.95 | 84.77 | 87.50 | 87.07 |
| **MAHA-Soft Logic** | **92.15** | **85.18** | **92.24** | **89.86** |

RadBot with CLIPSeg outperforms the DeepLabV3+ variant, especially on fine-grained subregions, highlighting the benefit of multimodal features. Table 3 studies threshold effects for both models.

We also performed an ablation study to evaluate the impact of different fusion strategies within MAHA. The results, presented in Table 4, report the segmentation performance of RadBot combined with each fusion strategy. It can be observed that the *Soft Logic* strategy consistently outperforms the other approaches, achieving a significant improvement in DSC on the BraTS 2020 dataset.

### 4.5 RadBot Mask Editor

While the segmentation masks generated by the model are significantly accurate, they may sometimes include false positives or false negatives, which can impact downstream analysis and clinical decision-making. To address this limitation, we introduce the **RadBot Mask Editor**, an interactive tool that allows radiologists or users to manually refine the segmentation masks. This feature ensures that the final segmentation aligns with clinical expectations and provides a higher degree of precision.

The RadBot Mask Editor enhances usability with adjustable parameters, including brush size control for precise or broad edits and a continuous painting mode for seamless modifications. A real-time overlay preview (Figure 5) allows users to visualize changes instantly, ensuring accurate segmentation refinement. Additionally, the editor supports coordinate-based editing, enabling precise pixel-level adjustments for fine-grained modifications. By integrating interactive editing capabilities, RadBot bridges the gap between automated segmentation and expert-driven refinement, ensuring clinically meaningful and highly accurate brain tumor analysis.

### 4.6 Interactive Responses of RadBot

RadBot extends beyond automated segmentation and survivability prediction by offering an interactive response system that enhances interpretability and clinical utility. Leveraging the capabilities of MMLLMs, RadBot can generate detailed, context-aware explanations and insights based on the segmented tumor regions and associated imaging data. As shown in Figure 6, RadBot is capable of generating informative and contextually relevant inquiry responses. These responses are not limited to predefined questions but also flexible to adapt to a variety of user inputs, enabling user to explore the data in a more interactive and meaningful way. RadBot combines the segmentation with interactive explanations, bridging automated analysis and expert interpretation. This establishes accurate, interpretable outputs, enabling users to make confident and informed decisions.

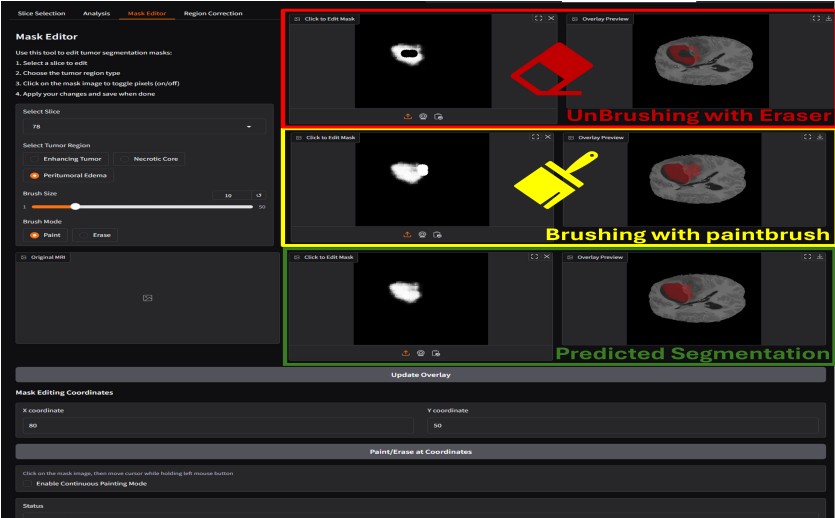

Figure 5: RadBot Mask Editor: manual refinement of segmentation masks to correct false positives/negatives. Yellow: brushing (add); red: unbrushing (remove). Adjustable brush size and continuous-painting mode enable precise edits.

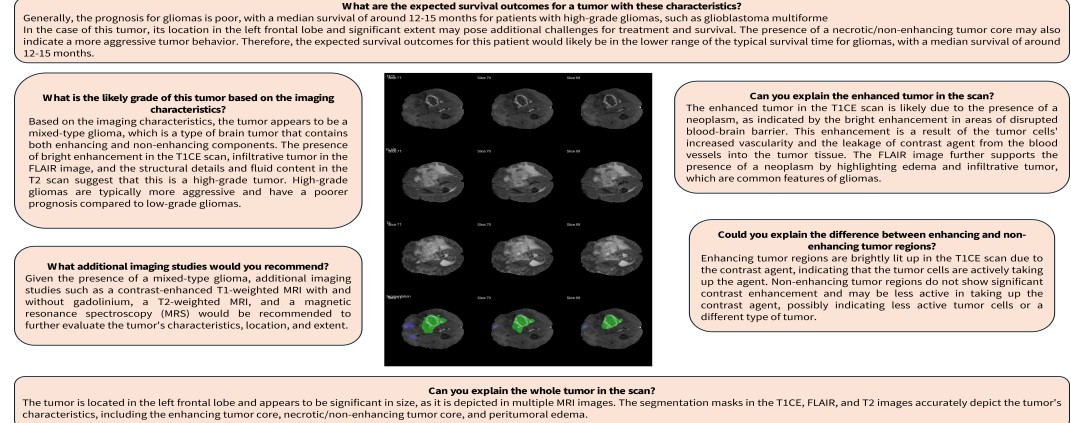

Figure 6: Interactive results of RadBot on tumor characteristics, inter-tumor differences, tumor grading, and survivability prediction.

## 5 CONCLUSION

In this work, we propose RadBot, a vision–language framework for brain tumor segmentation, interpretative analysis, and survivability prediction. RadBot integrates CLIPSeg and LLaVA: CLIPSeg leverages a frozen CLIP encoder for multimodal segmentation, and LLaVA contributes interactive insights. Evaluations on BraTS 2020/2021 validate accurate segmentations, clinically relevant interpretations, and robust survival predictions. A key contribution is the *RadBot Mask Editor* for interactive refinement. Further, we also proposed model agnostic hybrid augmentation (MAHA), an inference-time prompt ensemble method for brain tumor analysis. LLaVA-based question answering improves interpretability for decision support. By combining multimodal imaging and clinical data, RadBot strengthens survivability prediction and bridges automated segmentation with clinical workflow.

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
