# Interactive Brain-Tumor Analysis with Model Agnostic Hybrid Augmentation

## A    Supplementary Material: Qualitative Results

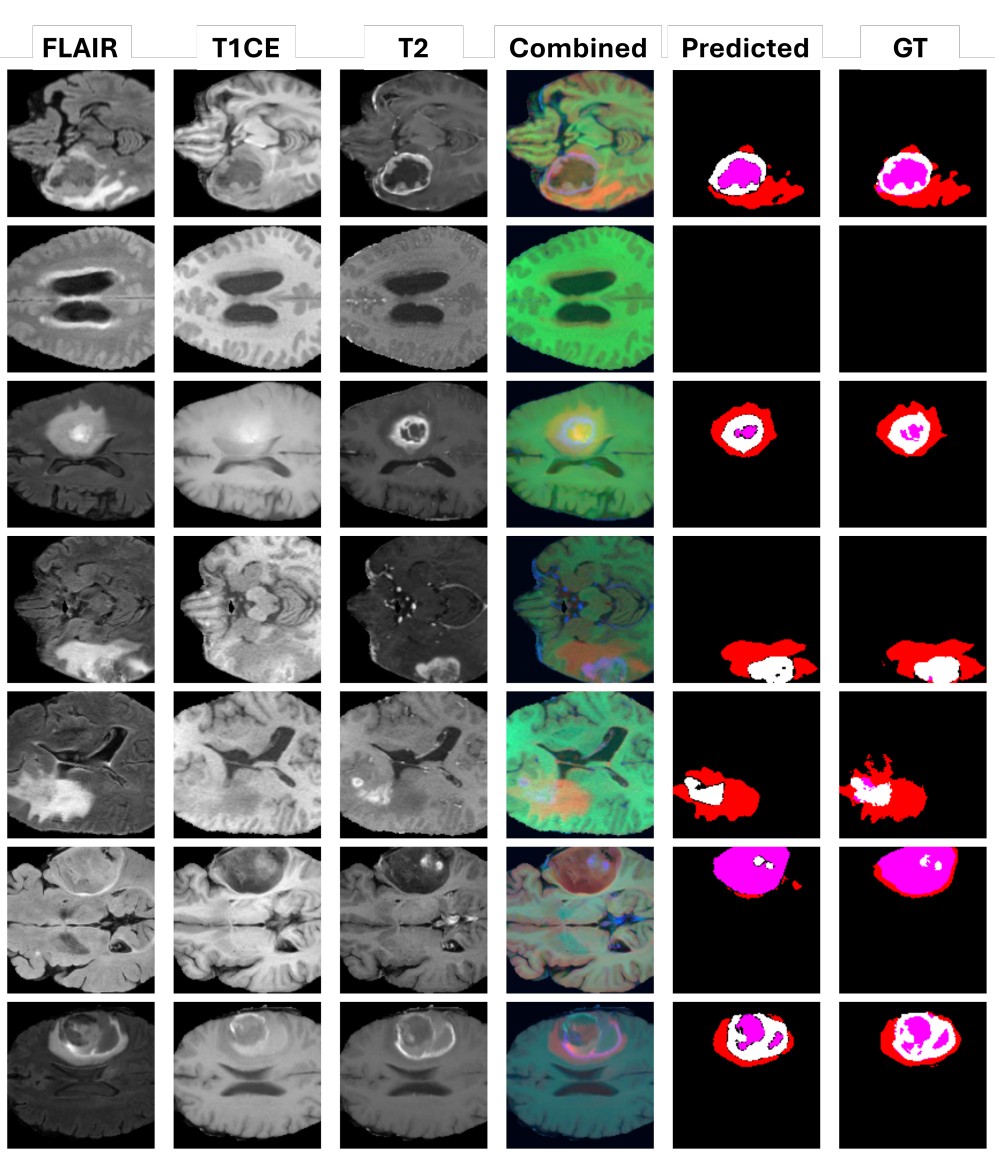

Figure 1: Brain tumor segmentation results for MRI scans from BraTS 2020 database for whole, core and enhance tumor region. Sample seven MRI scans (FLAIR, T1CE, T2 and combined) and GT (ground truth) from BraTS database and their respective predicted results using the proposed RadBot. Note: Red, White, Pink color shows the whole, enhance and core tumor region respectively.

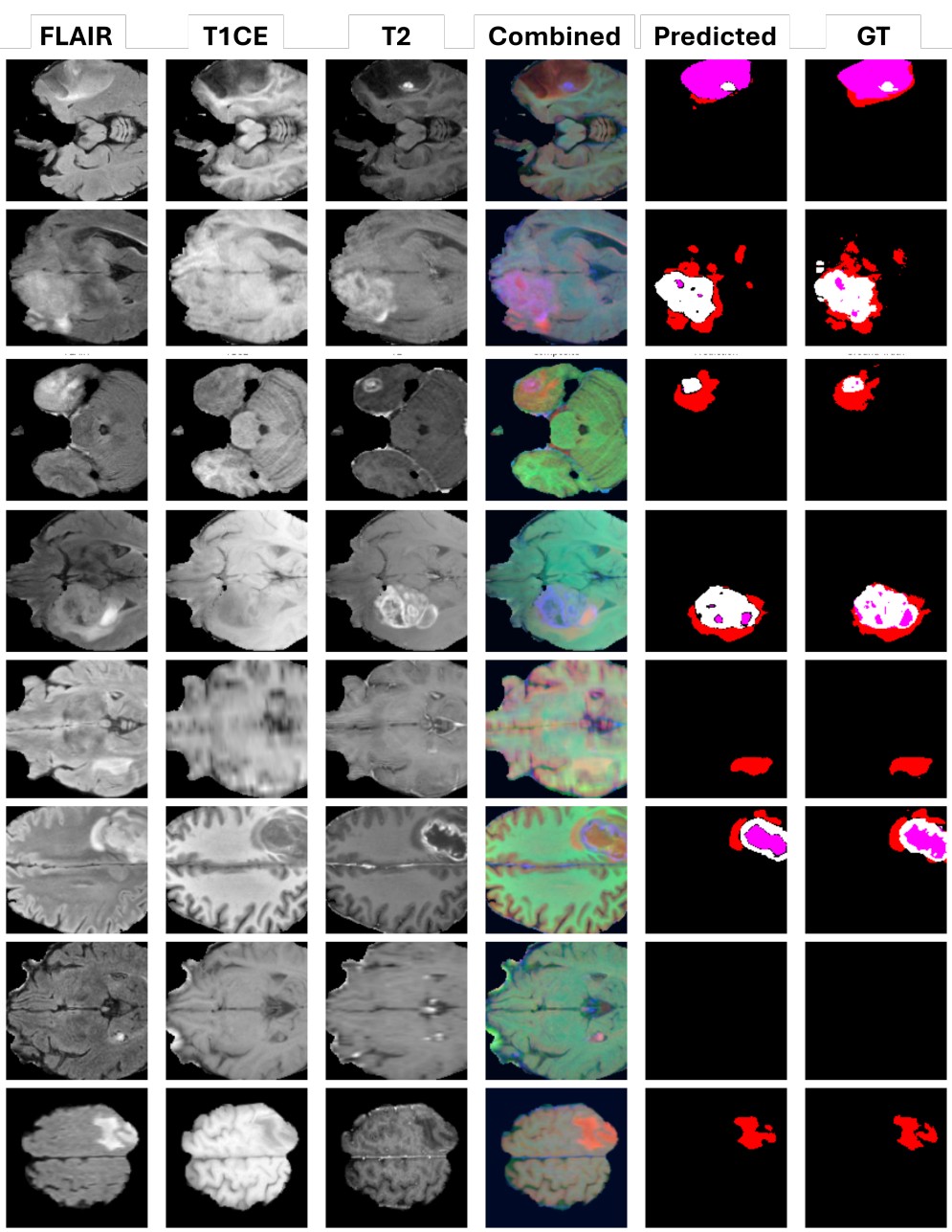

Figure 2: Brain tumor segmentation results for MRI scans from BraTS 2020 database for whole, core and enhance tumor region. Sample seven MRI scans (FLAIR, T1CE, T2 and combined) and GT (ground truth) from BraTS database and their respective predicted results using the proposed RadBot. Note: Red, White, Pink color shows the whole, enhance and core tumor region respectively.