# OpenReview forum: "Interactive Brain-Tumor Analysis with Model Agnostic Hybrid Augmentation"
_ICLR.cc/2026/Conference — ICLR 2026 Conference Withdrawn Submission_

### Official Review · Reviewer_py2q · 2025-10-19

**Soundness:** 3
**Presentation:** 1
**Contribution:** 3
**Rating:** 2
**Confidence:** 5

**Summary:**

This paper proposes RadBot, a vision–language framework for interactive brain tumor analysis. RadBot integrates CLIPSeg for MRI-based tumor segmentation and LLaVA for multimodal interpretative analysis. To address prompt sensitivity, the authors introduce Model-Agnostic Hybrid Augmentation (MAHA), an inference-time prompt ensembling strategy that fuses multiple linguistic prompts via adaptive thresholding to enhance segmentation robustness without retraining. Experiments on BraTS 2020 and 2021 demonstrate good performance compared to conventional CNN and transformer-based segmentation baselines.

**Strengths:**

1. Leveraging the LLM is interesting for medical applications.

2. The work addresses the prompt sensitivity problem, a major challenge in adapting general-purpose VL models to medical imaging.

3. Good performance.

**Weaknesses:**

The main contribution is not clearly stated. While the experiments center on segmentation, much of the discussion emphasizes prompt sensitivity, iterative interaction, and textual description, leading to ambiguity in the paper’s central technical focus.

The textual interpretability component (e.g., tumor description via LLaVA) lacks quantitative evaluation, making it unclear how these descriptive outputs contribute to clinical decision-making.

The rationale for choosing FLAIR, T1CE, and T2 modalities should be carefully justified, and the domain gap between natural RGB images (used in CLIP pretraining) and grayscale MRI inputs remains insufficiently discussed. The composed RGB-like visualization in Fig. 4 does not provide meaningful diagnostic information.

Table 3 shows that the threshold is quite sensitive to the segmentation performance.

The figures and visual presentation could be largely improved; key qualitative examples (e.g., segmentation overlays, interactive results) are visually unclear and lack contrast, which weakens the overall readability and impact.

**Questions:**

See the above weaknesses. Overall, this work requires substantial improvement.

---

### Official Review · Reviewer_SWoR · 2025-10-31

**Soundness:** 2
**Presentation:** 1
**Contribution:** 2
**Rating:** 0
**Confidence:** 5

**Summary:**

This paper introduces RadBot, an interactive framework for brain tumor analysis that combines a CLIP-based segmentation model with a language model to allow users to refine tumor segmentations and ask questions about the results. The work is evaluated on the BraTS dataset to segment and refine brain tumors.

**Strengths:**

The paper is good in terms of presenting a new framework which might be helpful for some researchers and clinicians. Overall, the strengths are:
1. User-Friendly, Interactive Framework
2. Simplistic design: They keep the model relatively lightweight which can be an advantage for deployment.
3. MAHA Prompt Ensembling seems an effective way to reduce prompt sensitivity and improve segmentation quality.

**Weaknesses:**

There are several limitations to this work. These are:
1. Custom BraTS Split: They used their own train-test split on the older BraTS datasets, which makes it harder to compare directly with other methods that used standard splits. The authors could have used the official BraTS splits or the newest challenge protocols to align with other published results.


2. Consequently to the previous point they are using an older BraTS versions. They’re using BraTS 2020 and 2021, but there are newer BraTS datasets in 2024 or 2025 that are larger and more diverse. I would use newer BraTS versions.


3. Missing T1 Modality: They only use three MRI modalities instead of all four that BraTS provides, skipping the T1 modality. It is not clear why.

4. The authors are using an older VLM (LLaVA). They rely on LLaVA, which is relatively older by 2025 standards. They could have used a more recent VLM like Qwen-VL or another state-of-the-art model to potentially improve performance.


5. Lack of Comparison to Interactive Baselines: The framework does not compare the approach to other recent interactive segmentation models like nnU-Net Interactive.


6. Semi-Automatic nature: Their method requires user interaction to refine the segmentation, so it’s not fully automatic. This is not a problem but I  would emphasize this is a feature for users who want interactivity, or explore making a fully automatic mode.

7. The paper do not provide standard deviations in their results tables, so it’s hard to see how consistent their performance is.

8. The state-of-the-art results they compare to, like nnU-Net’s older performance, may not reflect the very latest leaderboard numbers from 2025 or 2024. I would update the comparison to the most recent BraTS leaderboard results to ensure they’re benchmarking against the true current state-of-the-art. Also I would test on additional tumor segmentation datasets to show how well their method generalizes beyond just BraTS.

9. The method is not novel from a technical standpoint and the author did not make it clear that the focus is not novelty but rather building an interactive framework. They could acknowledge that and position their work as a practical framework rather than a novel technical advance.

**Questions:**

Please address the issues raised in the weaknesses section.

---

### Official Review · Reviewer_qYUW · 2025-11-01

**Soundness:** 2
**Presentation:** 3
**Contribution:** 3
**Rating:** 4
**Confidence:** 3

**Summary:**

This paper presents RadBot, an interactive vision-language framework for brain-tumor segmentation, interpretative analysis, and survivability prediction. RadBot integrates CLIPSeg for MRI-based vision-language segmentation and LLaVA for multimodal reasoning, visual-question-answering (VQA), and prognosis. To address prompt brittleness in VLMs, the authors propose Model-Agnostic Hybrid Augmentation (MAHA), an inference-time ensemble of multiple natural-language prompts fused via four strategies (mean, max, product, and soft-logic). A built-in RadBot Mask Editor further allows human experts to refine segmentation interactively. Experiments on BraTS 2020 and 2021 demonstrate state-of-the-art Dice scores (WT = 92.15, TC = 92.24, ET = 85.18), surpassing nnU-Net, TransBTS, MedNeXt, and U-Mamba. Overall, RadBot establishes a robust, interpretable, and user-in-the-loop pipeline for neuro-oncology imaging.

**Strengths:**

- RadBot couples segmentation, interpretive VQA, and survival analysis within one pipeline. The combination of a frozen CLIPSeg encoder, LLaVA-based reasoning, and LoRA fine-tuning for prognosis is well-engineered and practically relevant.
- MAHA effectively mitigates prompt-sensitivity without retraining. The soft-logic fusion rule is simple, generalizable, and yields measurable Dice improvements across tumor regions.
- Strong quantitative results on BraTS 2020/2021 with ablations (Tables 1–4) convincingly isolate MAHA’s contribution.
Visualizations reinforce interpretability and reproducibility.
- The Mask Editor and LLaVA Q&A components concretely realize interactive refinement and explainability—aligning with clinical workflow requirements.
- The paper is well-organized with lucid figures (Figs. 1–3), explicit hyperparameters, and reproducible implementation details (frozen encoder, BCE loss, LoRA).

**Weaknesses:**

- The framework primarily integrates existing models (CLIPSeg, LLaVA) with minimal architectural changes. Its originality lies in its system design rather than new learning theories or network architectures.
- The discussion omits recent benchmarks such as VILA-M3 (CVPR 2025), MSegNet (2025), and SAM-based medical segmentation (2024–25). Addressing these would clarify RadBot’s position among foundation-model adaptations.
- The paper lacks confidence intervals or variance metrics—important for clinical deployment claims.
- User-study or quantitative trust metrics for the Mask Editor / LLaVA interaction are missing.
- The survivability-prediction component lacks standard quantitative metrics (C-index, AUC, MAE) and baselines against CNN-based radiomics models.
- Issues such as bias propagation, hallucination in LLM outputs, and patient-data governance are not discussed.

**Questions:**

1. MAHA Design and Efficiency:
- How were the MAHA prompt variants created—manually curated or generated automatically (e.g., via paraphrasing models)?
- How sensitive is segmentation performance to the number and diversity of prompts, and what is the inference-time overhead compared to single-prompt inference?
- Can the method meet real-time constraints expected in clinical workflows?

2. Interpretability and Validation:
- Can the interpretability gains be quantified—e.g., through user studies, expert agreement metrics, or trust/time-to-correction analyses?

3. Prognostic and Generalization Performance:
- Please provide quantitative survivability-prediction results (C-index, AUC, MAE) and compare them with CNN- or transformer-based radiomics baselines.
- Have you tested cross-dataset generalization (e.g., TCIA, LGG, ISLES)? If not, what generalization mechanisms or domain-adaptation strategies are anticipated?

---

### Official Review · Reviewer_esrb · 2025-11-02

**Soundness:** 3
**Presentation:** 3
**Contribution:** 2
**Rating:** 4
**Confidence:** 4

**Summary:**

This paper presents **RadBot**, an interactive framework for **brain tumor segmentation, interpretative analysis, and survivability prediction**.
The system integrates three main components:
1. **CLIPSeg-based vision–language model (VLM)** for multimodal tumor segmentation.
2. **LLaVA-based multimodal LLM** for clinical interpretation and Q&A-style insights.
3. **Model-Agnostic Hybrid Augmentation (MAHA)** — a *training-free, inference-time prompt ensemble* that fuses multiple textual instructions via adaptive logit-level fusion (mean, max, product, soft logic) to mitigate prompt sensitivity.

Additionally, the **RadBot Mask Editor** provides an interactive GUI for radiologists to manually refine segmentation results.
On BraTS 2020 and 2021 benchmarks, RadBot+MAHA achieves **mean Dice ≈ 89.9**, surpassing classical CNN and transformer baselines (nnU-Net, H2NF-Net, SwinUNETR) without retraining.
The framework aims to bridge automated segmentation with clinically interpretable, user-in-the-loop analysis.

**Strengths:**

**Innovative multimodal integration:** Combines segmentation, interpretation, and prognosis in a single deployable framework, connecting VLMs (CLIPSeg) and MMLLMs (LLaVA) coherently.
- **Training-free augmentation (MAHA):** A clean, inference-time ensemble improving robustness without additional cost or retraining, practically beneficial for clinical deployment.
- **User interactivity:** The RadBot Mask Editor adds genuine usability for radiologists, offering visual corrections and interpretability rarely addressed in medical AI systems.

**Weaknesses:**

## Major
- **Limited methodological depth.** MAHA, while effective, is a straightforward logit-fusion ensemble lacking theoretical analysis or uncertainty modeling; novelty is modest.
- **Evaluation confined to BraTS datasets.** No tests on external or multi-institutional datasets, raising questions about generalization and clinical reliability.
- **Lack of quantitative interpretability evaluation.** The paper shows qualitative LLaVA outputs but does not assess factual correctness or alignment with clinical reports.
- **No baseline comparison for interactive editing.** The claimed benefit of the RadBot Editor remains anecdotal; no measurable improvement (e.g., post-edit Dice) is provided.

## Minor
- **Ablation studies focus on segmentation only.** Survival prediction lacks quantitative comparison with existing prognostic models.
- **System-level reproducibility.** Implementation details of GUI and hardware integration are underexplained, which matters for deployment feasibility.
- **Prompt set transparency.** MAHA’s prompt diversity (e.g., how many linguistic variants, selection process) is not systematically defined.

**Questions:**

1. How does MAHA perform under domain shifts (different MRI scanners, unseen datasets)?
2. Can LLaVA-based reasoning be quantitatively validated, e.g., with radiologist-rated correctness or factual grounding?
3. Is the RadBot Mask Editor integrated into the training feedback loop (e.g., human-in-the-loop fine-tuning)?
4. How many prompts were used in MAHA, and how sensitive are results to prompt count or linguistic diversity?
5. Does the survivability prediction branch meaningfully benefit from segmentation quality improvements, or are they independent modules?

---

### Note · Authors · 2025-11-13

I have read and agree with the venue's withdrawal policy on behalf of myself and my co-authors.